I fondi personali sono "complessi organici di materiali editi e/o inediti raccolti e/o prodotti da persone significative del mondo della cultura, delle professioni e delle arti prevalentemente dalla seconda metà del XIX secolo in poi"[1].

Anche se tali complessi si declinano in tipologie documentarie differenti (biblioteche d'autore, archivi di persona, archivi culturali), l'elemento aggregatore rimane l'individuo e dunque il corpus è documento e testimone degli interessi, delle attività e delle relazioni della persona nel contesto storico e culturale in cui ha operato[2].

La maggiore criticità nella gestione di tali complessi è la descrizione catalografica, a causa delle tipologie differenti di documenti e oggetti da descrivere, il che presuppone l'utilizzo di standard specifici per ciascuna tipologia di documento.

Gli esperimenti portati avanti per testare la validità di Wikidata per la descrizione analitica, esemplare per esemplare, di tali fondi hanno portato alla conclusione che non è opportuno usare Wikidata in tal senso. Migliore risulta essere infatti l'uso di un'istanza Wikibase adattata a questo scopo.

Tuttavia Wikidata è molto efficace nell'attività di descrizione di un fondo personale trattato nel suo insieme.

Descrivere un fondo personale in Wikidata permette di inserire il fondo in una rete di relazioni i cui nodi sono rappresentati dal soggetto produttore, dal soggetto conservatore, dai precedenti possessori, dal luogo di conservazione, dagli autori di note, dediche e postille rintracciate nei volumi ecc. Tali relazioni varcano i confini di interesse dell'ente conservatore per ripristinare connessioni andate perdute per varie cause, tra cui lo smembramento dei fondi personali e la loro conservazione presso enti culturali diversi.

Altro vantaggio è la possibilità di aumentare la conoscenza sul materiale documentario e sul suo possessore, grazie ai dati provenienti da fonti di informazioni differenti, che convergono in Wikidata.

---

[1] Commissione nazionale biblioteche speciali, archivi e biblioteche d'autore, *Linee guida sul trattamento dei fondi personali*, versione 15.1, 31 marzo 2019, p.1.
<https://www.aib.it/documenti/linee-guida-sul-trattamento-dei-fondi-personali/>
[2] Ibidem

Dalle ricerche effettuate risulta che, pur essendo molti i fondi descritti in Wikidata[3], tuttavia tali item utilizzano un data model non standardizzato e un "vocabolario" dei termini usati per le etichette non controllato. Questa disomogeneità provoca una grande dispersione dei dati e la difficoltà nel rintracciare tutti gli elementi che si riferiscono alla tipologia di fondo personale.

La creazione di un Wikidata:Wikiproject dedicato potrebbe risolvere tale problematica, fungendo da punto di raccordo e fonte di buone pratiche per quanti desiderano inserire un fondo personale in Wikidata.

La relazione proporrà la creazione di tale Wikiproject e un data model "fondo personale", che andrà a ricalcare le informazioni presenti nella scheda di rilevazione fondi a cui la Commissione nazionale Biblioteche speciali, archivi e biblioteche d'autore AIB sta lavorando e che sarà resa disponibile ai colleghi bibliotecari nel corso del 2025.

In questo modo le due attività e la loro comunicazione potrebbero camminare parallelamente riuscendo ad intercettare più professionisti delle istituzioni GLAM sia nell'uso della scheda rilevazione fondo personale all'interno della propria istituzione, sia nella corrispondente creazione di un item in Wikidata relativo al fondo personale a cui si sta lavorando.

Gli scopi del progetto saranno:

- l'inserimento, arricchimento e valorizzazione di dati sui fondi personali in Wikidata;
- l'implementazione e il mantenimento di ontologie e thesauri multilingue relativi alla descrizione dei fondi personali;
- l'interconnessione tra i cataloghi delle collezioni e Wikidata;
- l'inclusione dei dati in Wikipedia e nei suoi progetti gemelli.

---

[3] In Wikidata non esiste un item "fondo personale", che possa essere usato come valore della proprietà "istanza di" per i fondi e complessi documentari corrispondenti a tale definizione.
Il termine "fondo personale" è stato usato finora solo nelle labels di descrizione di alcuni item. Esiste invece il valore "biblioteca personale".
Dalle query effettuate risultano 10 item di "biblioteca personale" (Q106402388); 386 sono gli item corrispondenti a "fondo speciale" (Q4431094). 465 sono gli items corrispondenti a "collezione bibliotecaria" (Q856592), tra i quali troviamo anche alcune biblioteche personali.
Per quanto riguarda gli archivi descritti in Wikidata invece si possono rilevare più di 100.000 elementi di "fondo archivistico", 19 items di "archivi di persona" (Q27032347), un unico elemento descritto come "archivio personale" (Q7170558). Gli elementi con istanza di "archivio di famiglia" (Q27032283) sono 18. Per gli archivi si veda il Wikiproject Archival Description (<https://www.wikidata.org/wiki/Wikidata:WikiProject_Archival_Description>)