# OpenReview forum: "Una proposta per la gestione dei fondi personali in Wikidata"
_wikimedia.it/Wikidata_and_Research/2025/Conference — WD&R Paper_

### Official Review · ~Elena_Marangoni1 · 2025-01-10
**Wikiproject per l'inserimento di fondi personali in Wikidata**

**Originality:** 5
**Impact:** 5
**Confidence:** 4

**Review:**

La proposta è di grande interesse perché riflette su un modello che metta in relazione la descrizione di fondi personali su Wikidate e uno strumento che sarà utilizzato nelle biblioteche italiane (la scheda di rilevazione fondi della Commissione nazionale Biblioteche speciali, archivi e biblioteche d’autore AIB). Si prende anche  in considerazione l'interrelazione  tra Wikibase e Wikidata (per la descrizione analitica dei fondi il primo  e del fondi personali nel loro insieme il secondo). La sperimentazione di un Wikiproject contibuirà ad arrichire gli strumenti disponibili per la comunità degli wikidatiani, in particolare per le istituzioni GLAM che potrranno adottare un percorso di lavoro gia testato.

**Compliance:**

5

**Notes:**

Revisione del testo finale di Silvia Bruni:
"Il testo descrive molto accuratamente i presupposti della sperimentazione e introduce con accuratezza il modello di lavoro predisposto su Wikidata. Assolutamente interessante è la strutturazione in Wikidata di un ambiente MAB, in cui le relazioni fra le informazioni sono collegabili indipendentemente dagli strumenti di catalogazione e descrizione adottati dalle istituzioni culturali che li conservano. Non ho rilievi da fare."

**Scientific Quality:**

5

---

### Official Review · ~Rossana_Morriello1 · 2025-01-11
**Proposta chiara e rilevante**

**Originality:** 4
**Impact:** 5
**Confidence:** 3

**Review:**

La proposta è coerente con il tema del convegno, vengono presentati con chiarezza gli obiettivi e la metodologia. Si propone di illustrare un progetto già avviato e un data model che verrà reso disponibile nel 2025, dunque effettivamente utile per i bibliotecari universitari e della ricerca che spesso si trovano a dover trattare fondi personali e d'autore.

**Compliance:**

5

**Scientific Quality:**

5

---

### Decision · Program_Chairs · 2025-02-05

Accept (Paper)